# Molecular Characterization of Sterol C4-Methyl Oxidase in *Leishmania major*

**DOI:** 10.3390/ijms252010908

**Published:** 2024-10-10

**Authors:** Yu Ning, Somrita Basu, Fong-fu Hsu, Mei Feng, Michael Zhuo Wang, Kai Zhang

**Affiliations:** 1Department of Biological Sciences, Texas Tech University, Lubbock, TX 79409, USA; yuning1208@gmail.com (Y.N.); sombasu@ucdavis.edu (S.B.); 2Mass Spectrometry Resource, Division of Endocrinology, Diabetes, Metabolism, and Lipid Research, Department of Internal Medicine, Washington University School of Medicine, St. Louis, MO 63110, USA; fhsu@wustl.edu; 3Department of Pharmaceutical Chemistry, School of Pharmacy, The University of Kansas, Lawrence, KS 66047, USA; m842f342@ku.edu (M.F.); michael.wang@ku.edu (M.Z.W.)

**Keywords:** CYP5122A1, ERG25, *Leishmania*, sterol, demethylase

## Abstract

Sterol biosynthesis requires the oxidative removal of two methyl groups from the C-4 position by sterol C-4-demethylase and one methyl group from the C-14 position by sterol C-14-demethylase. In *Leishmania donovani*, a CYP5122A1 (Cytochrome P450 family 5122A1) protein was recently identified as the bona fide sterol C-4 methyl oxidase catalyzing the initial steps of C-4-demethylation. Besides CYP5122A1, *Leishmania* parasites possess orthologs to ERG25 (ergosterol pathway gene 25), the canonical sterol C-4 methyl oxidase in *Saccharomyces cerevisiae*. To determine the contribution of *CYP5122A1* and *ERG25* in sterol biosynthesis, we assessed the essentiality of these genes in *Leishmania major*, which causes cutaneous leishmaniasis. Like in *L. donovani*, *CYP5122A1* in *L. major* could only be deleted in the presence of a complementing episome. Even with strong negative selection, *L. major* chromosomal *CYP5122A1*-null mutants retained the complementing episome in both promastigote and amastigote stages, demonstrating its essentiality. In contrast, the *L. major ERG25*-null mutants were fully viable and replicative in culture and virulent in mice. Deletion and overexpression of ERG25 did not affect the sterol composition, indicating that ERG25 is not required for C-4-demethylation. These findings suggest that CYP5122A1 is the dominant and possibly only sterol C-4 methyl oxidase in *Leishmania*, and inhibitors of CYP5122A1 may have strong therapeutic potential against multiple *Leishmania* species.

## 1. Introduction

*Leishmania* parasites are trypanosomatid protozoans transmitted through the bite of sandflies. During their life cycle, these protozoans alternate between extracellular flagellated promastigotes in the sandfly gut and intracellular non-flagellated amastigotes in mammalian macrophages. *Leishmania* parasites cause a group of diseases in humans with symptoms ranging from self-limiting skin lesions (cutaneous leishmaniasis) to life-threatening infections of the bone marrow, liver, and spleen (visceral leishmaniasis) [1]. Current treatments are plagued with high toxicity, low efficacy, and increasing resistance [2]. Thus, new drugs are needed to control leishmaniasis.

The sterol biosynthesis pathway presents multiple targets for antileishmanial drugs. Among them is sterol C14-demethylase (C14DM, also known as CYP51), which is the primary target of azoles [3,4]. C14DM is a heme-containing enzyme catalyzing the oxidative removal of the C14-methyl group from sterol intermediates (Figure 1) [5]. Genetic or chemical inhibition of C14DM in *Leishmania* leads to increased plasma membrane fluidity, hypersensitivity to heat and compromised mitochondrial functions [6,7]. In addition to C14-demethylation, two more methyl groups need to be removed from the C4 position during sterol synthesis (Figure 1). In *Saccharomyces cerevisiae*, three enzymes catalyze sterol C4-demethylation: an O_2_-dependent C4 sterol methyl oxidase (ERG25) that sequentially generates hydroxy, aldehyde and carboxylate intermediates; a C4 decarboxylase (ERG26) removing CO_2_ from carboxylate intermediates to yield 3-ketosterol intermediates; and a 3-ketosterol reductase (ERG27), which regenerates the C3-hydroxy group [8,9,10]. While *S. cerevisiae* and vertebrates use the same C4 sterol methyl oxidase (ERG25) on both C4 methyl groups, plants have two distinct ERG25 homologs that oxidize 4,4-dimethylsterols and 4α-methylsterols, respectively [11,12] (Figure 1).

A recent study has identified *Leishmania donovani* CYP5122A1 (XP_003861867.1) as a non-canonical sterol C4-methyl oxidase [13]. CYP5122A1 is well conserved among multiple trypanosomatid species. Recombinant LdCYP5122A1 protein binds and converts C4-methylated sterols into oxidized metabolites in the presence of NADPH and oxygen. Importantly, CYP5122A1 is essential for both promastigotes and intracellular amastigotes in *L. donovani*. Heterozygous *CYP5122A1* knockout in which one of the two *CYP5122A1* alleles is deleted shows reduced infectivity in mice [13,14]. In addition, inhibitors of both CYP5122A1 and C14DM display superior antileishmanial activity against *L. donovani* in comparison to C14DM-selective inhibitors [13,15]. These findings suggest that C4DM is a promising drug target.

In addition to CYP5122A1, *Leishmania* parasites possess a putative *ERG25* in their genomes. It is not known whether the leishmanial ERG25 protein is involved in the C4-demethylation reaction or whether its function overlaps with CYP5122A1. Amongst trypanosomatids, ERG25 orthologs are present in *Leishmania* spp., *Angomonas deanei*, *Bodo saltans*, *Crithidia fasciculata*, *Endotrypanum monterogeii*, *Leptomonas* spp., *Paratrypanosoma confusum* and *Porcisia hertigi*, but absent in *Trypanosoma* spp. [16]. In this study, we investigated the roles of ERG25 in growth, virulence and sterol synthesis by characterizing genetic mutants of *ERG25* in *Leishmania major*. We also explored the essentiality of CYP5122A1 in *L. major*. Our findings argue that CYP5122A1 is the sole sterol C4-methyl oxidase in *Leishmania*.

## 2. Results

### 2.1. Identification and Cellular Localization of ERG25 and CYP5122A1 (22A1) in L. major

We recently characterized the *Leishmania donovani* CYP5122A1 protein (TriTrypDB: LdBPK_270090.1; GenBank: XP_003861867.1) as a sterol C4-methyl oxidase catalyzing the sequential oxidation of lanosterol to form C4-oxidation metabolites [13] (Figure 1). *L. major* CYP5122A1 (TriTrypDB: LmjF.27.0090; GenBank: XP_003721815.1) is 94.4% identical to *L. donovani* CYP5122A1 with 606 amino acids and one transmembrane domain (AA51-74) (Appendix A). In addition to CYP5122A1 (referred to as 22A1 in this study), the *L. major* genome contains an ortholog of the *Saccharomyces cerevisiae* ERG25 gene (TriTrypDB: LmjF.36.2540), which encodes the sterol C4-methyl oxidase in yeast. *L. major* ERG25 has 402 amino acids with two transmembrane domains (AA67-90 and 136-159) showing a modest 26% identity to ScERG25 from AA194-384 (Appendix A). Based on information from TriTrypDB, CYP5122A1 orthologs are present in multiple *Leishmania* and *Trypanosoma* species, but ERG25 orthologs are not found among *Trypanosoma* spp.

To determine the cellular localization of 22A1, a 22A1-GFP fusion protein gene was introduced into *L. major* wild-type (WT) parasites. The resulting WT +22A1-GFP promastigotes were labeled with an anti-*T. brucei* BiP antibody to mark the endoplasmic reticulum (ER) [17]. Imaging analysis showed a ~72% overlap between 22A1-GFP and BiP (Figure 2A). Similar observations were obtained when a GFP-ERG25 fusion protein was expressed in *L. major* WT parasites (Figure 2B; ~80% overlap with BiP). Both GFP-ERG25 and 22A1-GFP showed the expected molecular weights by Western bot (Figure 2C). These findings indicate that 22A1 and ERG25 are primarily located in the ER.

### 2.2. Successful Generation of ERG25-Null but Not 22A1-Null Mutant in L. major

Using the targeted gene deletion approach based on homologous recombination, we replaced the two endogenous alleles of *ERG25* with drug-resistant genes for puromycin (*PAC*) and hygromycin (*HYG*) [18]. The resulting *ERG25*^+/−^ (heterozygous knockout) and *erg25*^−/−^ (homozygous knockout) clones were verified by Southern blot (Figure 3A and Appendix A). Expression of *ERG25* was restored when a pXG-ERG25 plasmid was introduced into *erg25*^−/−^ to generate the add-back strain *erg25*^−/−^ +*ERG25*.

With the same homologous recombination approach, we generated the heterozygous *22A1*^+/−^ knockout. However, repeated attempts to generate the *22A1*^−/−^ null mutant were unsuccessful, which was reminiscent of CYP5122A1 in *L. donovani* [13]. To delete the chromosomal *22A1* alleles, the pXNG4-22A1 plasmid was introduced into *22A1*^+/−^ parasites. In the presence of pXNG4-22A1, we were able to remove the second chromosomal *22A1* to generate *22A1*^−/−^ +pXNG4-22A1 (Figure 3B and Appendix A).

### 2.3. ERG25 Is Not Required for the Growth or Virulence of L. major Promastigotes

As shown in Figure 4A, *erg25*^−/−^ and *erg25*^−/−^ +*ERG25* (add-back) promastigotes replicated at the same rate as WT parasites during the log phase. After entering the stationary stage, *erg25*^−/−^ had similar percentages of round cells and dead cells as WT and add-back parasites (Figure 4B,C). Thus, deletion or episomal overexpression of *ERG25* does not affect the survival, morphology or proliferation of *L. major* promastigotes in culture.

To investigate the virulence of *erg25*^−/−^ in a mouse model, late stationary phase promastigotes were injected into the footpads of female BALB/c mice. As shown in Figure 4D, mice infected by *erg25*^−/−^ or *erg25*^−/−^ +*ERG25* developed lesions at similar rates as WT-infected mice. *ERG25* mutants and add-back appeared to have a slower growth rate in mice than WT based on parasite number analyses at 4 and 7 weeks post-infection, although only one mouse was used for each group (Figure 4E). These findings suggest that while ERG25 is not essential for virulence in *L. major*, endogenous ERG25 expression is needed for optimal growth in mice. In addition, the abundance and cellular localization of LPG and GP63, two major virulence factors in *Leishmania*, appeared to be unaltered in *erg25*^−/−^ by immunofluorescence microscopy analysis (Appendix A).

### 2.4. ERG25 Deletion or Add-Back Does Not Affect Sterol Composition in L. major

To determine if ERG25 is involved in sterol biosynthesis, we analyzed the sterol contents in log-phase promastigotes of WT and *erg25^−^* by electron ionization (EI) gas chromatography–mass spectrometry (GC-MS) [6,19]. We also examined the sterols in *erg25*^−/−^ +*ERG25* and *erg25*^−/−^ +*GFP-ERG25* parasites, which overexpress ERG25 or GFP-ERG25 proteins from high-copy number plasmids [20]. As previously reported, the four major sterol species in *L. major* WT promastigotes consisted of 5-dehydroepisterol, ergosterol, cholesta-5,7,24-trienol and cholesterol, identified based on molecular weights, retention times and EI spectra (Figure 5A) [6]. Similar sterol compositions (27–31% ergosterol, 42–51% 5-dehydroepisterol, 5–7% cholesterol and 15–24% cholesta-5,7,24-trienol) were detected in *erg25*^−/−^, *erg25*^−/−^ +*ERG25* and *erg25*^−/−^ +*GFP-ERG25* parasites (Figure 5B–E). Furthermore, the cellular levels of major sterols in *erg25*^−/−^ were similar to WT and add-back parasites based on a comparison of endogenous sterol peaks to that of the internal standard cholesta-3,5-diene (retention time: 8.78) (Figure 5A–D). Overall, these results indicate that *ERG25* is not required for sterol synthesis in *L. major*.

### 2.5. 22A1 Is Indispensable during the Promastigote Stage of L. major

To assess the essentiality of 22A1 in culture promastigotes, we tested whether pXNG4-*22A1* can be removed from the *22A1*^−/−^ +pXNG4-*22A1* parasites. The presence of a thymidine kinase gene on pXNG4-*22A1* made these parasites susceptible to negative selection by ganciclovir (GCV). With GCV treatment, parasites tend to lose pXNG4-*22A1* if the plasmid is not essential [13,21]. When *22A1*^+/−^ +pXNG4-*22A1* and *22A1*^−/−^ +pXNG4-*22A1* parasites were cultivated in the presence of nourseothricin (SAT), 80–85% of them had high GFP expression indicating a high pXNG4-*22A1* level (Figure 6A and Appendix A). Without SAT, *22A1*^+/−^ +pXNG4-*22A1* promastigotes (which had one chromosomal *22A1* allele) gradually lost the plasmid as the GFP-high cells decreased from ~80% to ~10% in six passages (Figure 6A). With GCV treatment, the plasmid loss was accelerated and near complete after 5–6 passages due to the negative selection (Figure 6A). Therefore, in the presence of endogenous *22A1*, the pXNG4-*22A1* episome is dispensable.

Notably, the GFP-high cells in *22A1*^−/−^ +pXNG4-*22A1* (which are chromosomal null for *22A1*) sustained at 79–83% in the absence of GCV and 59–64% in the presence of GCV after six passages (Figure 6A,B), suggesting that these promastigotes could not lose the episome even with negative selection. When single cells (clones) were isolated from the *22A1*^−/−^ +pXNG4-*22A1* GCV population (Figure 6C from GFP-low and Figure 6D from GFP-high) and allowed to proliferate, they showed a mixture of GFP-high (51–65%) cells and GFP-low (35–49%) cells similar to the original population, suggesting that the pXNG4-*22A1* plasmid cannot be eliminated (Figure 6B–D). Together, these findings indicate that 22A1 is required for the survival and replication of *L. major* promastigotes.

Additionally, we compared the growth rates of *22A1*^+/−^ +pXNG4-*22A1* and *22A1*^−/−^ +pXNG4-*22A1* with WT promastigotes (Figure 6E). When cultivated in the absence of GCV, these parasites proliferated at very similar rates. However, in the presence of GCV, *22A1*^−/−^ +pXNG4-*22A1* showed a significant growth delay in comparison to WT and *22A1*^+/−^ +pXNG4-*22A1* cells at day 1–2 post-inoculation (Figure 6E, two clones of *22A1*^−/−^ +pXNG4-*22A1* were tested). It is possible that the GCV-induced cytotoxicity affected *22A1*^−/−^ +pXNG4-*22A1* parasites more due to their inability to lose the plasmid.

### 2.6. 22A1 Is Essential for L. major during the Amastigote Stage

To determine if 22A1 is required during the amastigote stage, stationary phase promastigotes were injected into the footpad of BALB/c mice. Half of the mice were treated with GCV daily for 14 days, while the other half received an equal volume of sterile PBS as controls. As indicated in Figure 7A,B, mice infected by WT and *22A1*^+/−^ showed similar lesion progression and had equivalent parasite numbers. Thus, losing one chromosomal *22A1* allele does not compromise *L. major* virulence or growth in mice. Also, GCV treatment did not affect footpad lesion development or parasite loads in WT- and *22A1*^+/−^-infected mice.

Mice infected by *22A1*^+/−^ +pXNG4-*22A1* or *22A1*^−/−^ +pXNG4-*22A1* and received PBS treatment also showed similar lesion progression and parasite numbers as WT-infected mice (Figure 7A,B). In contrast, GCV treatment significantly delayed lesion development and amastigote replication in these mice. This observation is in line with the effect of GCV on those parasites in culture (Figure 6E). The reduced growth of *22A1*^+/−^ +pXNG4-*22A1* + GCV (four weeks delay in comparison to WT) may be due to the cytotoxic effect of GCV during the initial phase post-infection when parasites contained a high level of pXNG4-*22A1* (Appendix A). The delay was much more pronounced with the *22A1*^−/−^ +pXNG4-*22A1* + GCV group (nine weeks), which could arise if these parasites were unable to lose the plasmid.

To examine the retention of pXNG4-*22A1*, we performed qPCR analyses on the genomic DNA extracted from lesion-derived amastigotes. As shown in Figure 7C, the plasmid copy number was 0.45–0.75 per cell for *22A1*^+/−^ +pXNG4-*22A1* amastigotes with PBS treatment at 5–7 weeks post-infection. With GCV treatment, less than 0.1 copy per cell was retained for *22A1*^+/−^ +pXNG4-*22A1* at 7–11 weeks post-infection. Meanwhile, the *22A1*^−/−^ +pXNG4-*22A1* amastigotes contained 1.3–2.2 copies per cell with PBS treatment and 0.8–1.8 copies per cell with GCV treatment, even after 16 weeks post-infection. Based on these findings, we conclude that 22A1 is essential for the survival of *L. major* in the amastigotes stage.

### 2.7. 22A1 Overexpression Confers Resistance to Posaconazole and DB766 but Does Not Affect Sterol Composition or Stress Response in L. major

In *L. donovani*, CYP5122A1 overexpression (from pXNG4-LdCYP5122A1) led to increased tolerance to stress conditions such as high temperature, acidic pH and starvation [13]. To test if the 22A1 had a similar impact in *L. major*, stationary phase promastigotes of *22A1*^+/−^ +pXNG4-*22A1* and *22A1*^−/−^ +pXNG4-*22A1* were exposed to 37 °C or pH 5.0 conditions and their tolerance was measured as previously described [6,13]. As summarized in Appendix A, 22A1 overexpressors showed a similar level of survival as WT and *22A1*^+/−^ parasites. We also examined whether 22A1 heterozygous knockout and overexpression alter the sterol composition in *L. major* by liquid chromatography–tandem mass spectrometry (LC-MS/MS). While the major sterol species were very similar between WT and *22A1* mutants (Appendix A), we detected a slight accumulation of squalene, a sterol biosynthesis intermediate, in *22A1*^+/−^ +pXNG4-*22A1* parasites (Appendix A). The reason for this minor accumulation is not clear. One possibility is that 22A1 overexpression causes increased metabolism of sterol biosynthesis intermediates leading to upregulation of squalene synthase and/or downregulation of squalene monooxygenase. Compared to *CPY5122A1* mutants in *L. donovani*, which showed altered levels of 4,14-methylated sterol intermediates [13], *22A1* overexpression and heterozygous knockout had little influence on sterol composition in *L. major*.

Sterol synthesis inhibitors such as ketoconazole, posaconazole and DB766 target C14DM and/or C4DM [6,13,22]. To test if *22A1* heterozygous knockout and overexpressors have altered sensitivity to these inhibitors, we cultivated promastigotes in various concentrations of ketoconazole (0–15 µM), posaconazole (0–6 µM) or DB766 (0–32 nM), and measured culture densities after 48 h. The effective concentrations to inhibit 50% of growth (referred to as EC50) were calculated using cells grown in the absence of drugs as controls (Table 1). As illustrated in Figure 8A, these parasites showed similar levels of susceptibility to ketoconazole except that at higher concentrations (10–15 µM), WT and *22A1*^+/−^ parasites were slightly more sensitive. For posaconazole and DB766, *22A1*^+/−^ +pXNG4-*22A1* and *22A1*^−/−^ +pXNG4-*22A1* were more resistant than WT and *22A1*^+/−^ parasites, as their EC50s were 1.5–3.0-fold higher (Figure 8B,C and Table 1). These findings suggest that 22A1 overexpression leads to modest resistance to C4DM inhibitors.

### 2.8. Genetic Manipulation of 22A1 or ERG25 Do Not Affect the Expression Level of C14DM

To determine if *22A1* heterozygous knockout or overexpression affects the expression of C14DM (CYP51), Western blots were conducted using antibodies against *L. donovani* CYP5122A1 and CYP51 (Figure 9). Clearly, genetic manipulation of *22A1* had little impact on C14DM (CYP51) expression at the protein level. Conversely, C14DM-null and add-back parasites contained similar levels of 22A1 as WT. As expected, 22A1 heterozygous knockout and overexpressors showed reduced and elevated levels of 22A1, respectively. Furthermore, deletion or overexpression of ERG25 did not affect the levels of 22A1 or C14DM (CYP51) (Figure 9). These findings indicate that expressions of 22A1, C14DM and ERG 25 are not linked.

## 3. Discussion

Sterol biosynthesis requires the formation of lanosterol, followed by the removal of three methyl groups: two from the C4 position and one from the C14 position (Figure 1). Compared to the C14DM/CYP51, which is well characterized biochemically and genetically as the main target for azole drugs, the C4DM is less studied. In *L. donovani* (the etiological agent for visceral leishmaniasis), we recently identified CYP5122A1 as a bona fide sterol C4-methyl oxidase capable of converting C4-methylated sterols into mono-hydroxy, aldehyde and carboxylate intermediates in vitro [13]. Chromosomal CYP5122A1-null mutants cannot survive without a complementing CYP5122A1-episome, indicating that it is required for *L. donovani* promastigotes and amastigotes. In this report, we investigated whether the essentiality of CYP5122A1 can be extended to *L. major*, which causes cutaneous leishmaniasis. Previous studies from us and others have revealed differences in the requirement of sterol and sphingolipid metabolism among *Leishmania* species [17,23,24,25]. For example, while C14DM/CYP51 can be deleted from *L. major*, it has been considered indispensable in *L. donovani* [17,24], although a recent study managed to generate viable homozygous knockout using the CRISPR-Cas9 gene-editing approach [26]. In addition, the SL-degrading enzyme ISCL is required for the virulence of *L. major* but not *L. amazonensis* [23,25]. Thus, it is important to determine if different *Leishmania* species have distinct requirements for C4DM.

Using the same plasmid segregation method described for *L. donovani* CYP5122A1 [13,21], we showed that *L. major 22A1*^−/−^ +pXNG4-*22A1* promastigotes retained a high level of episome even when facing strong negative selection from GCV treatment, whereas *22A1*^+/−^ +pXNG4-*22A1* lost the episome after six passages. When single clones isolated from *22A1*^−/−^ +pXNG4-*22A1* GCV treatment group were allowed to proliferate, they showed a mix of GFP-low and GFP-high cells, suggesting that GFP-low cells (lacking pXNG4-*22A1*) could not perpetuate by themselves. Thus, like in *L. donovani*, 22A1 is essential for *L. major* promastigotes. Given the fact that *L. major* C14DM/CYP51-null mutants are viable [6], these findings suggest that the removal of the two C4-methyl groups is more crucial than the removal of the C14-methyl group. It is possible that the accumulation of C4-methyl intermediates is more harmful to *Leishmania* membrane than the accumulation of C14-methyl intermediates [27].

In the intracellular mammalian stage, *L. major* amastigotes are less dependent on de novo lipid synthesis and can salvage and remodel host lipids to support their activity [6,28,29,30]. To determine whether 22A1 is dispensable for amastigotes, BALB/c mice infected with *22A1*^−/−^ +pXNG4-*22A1* were treated with GCV or PBS for 14 days. As illustrated in Figure 7, GCV treatment dramatically reduced the virulence and growth of *22A1*^−/−^ +pXNG4-*22A1* in mice; yet, these parasites retained the pXNG4-*22A1* plasmid at 16 weeks post-infection. These results are consistent with the notion that 22A1 is required during the amastigote stage. For mice infected by *22A1*^+/−^ +pXNG4-*22A1*, GCV treatment caused an initial delay in footpad lesion formation, which might arise from the elevated level of pXNG4-*22A1* in the early stage of infection. After 7 weeks, these parasites lost the episome and displayed normal levels of virulence and growth in mice, indicating that the plasmid is not essential for these parasites, which contain one of chromosomal *22A1*. These findings are largely consistent with the data from *L. donovani* CYP5122A1. However, one distinction is that *L. major 22A1*^+/−^ (heterozygous knockout) shows a WT level of infectivity while *L. donovani* heterozygous knockout has attenuated virulence [13,14]. Also, *L. major 22A1* overexpression and heterozygous knockout did not affect stress response or bulk sterol composition. These results suggest that *L. major* parasites can tolerate the impact of 22A1 over- or under-expression better than *L. donovani*. We did detect a slight accumulation of squalene in 22A1-overexpressing cells, suggesting that its overproduction can affect the early steps of sterol synthesis.

Overall, our results demonstrate that 22A1 is essential for *L. major* amastigotes, just like it is for *L. donovani* amastigotes. This is notable because most sterols in intracellular amastigotes consist of host-derived cholesterol and only a small portion was synthesized de novo [6,31]. These findings raise the potential of 22A1 as a pan-*Leishmania* drug target. In culture promastigotes, the overexpression of 22A1 led to 2–3-fold resistance to several 22A1 and CYP51 inhibitors including DB766 and posaconazole (Figure 8). Future studies will aim to identify more potent and selective inhibitors of 22A1.

Despite these findings, it is not clear whether CYP5122A1 is the sole sterol C4 methyl oxidase in *Leishmania*. In *S. cerevisiae* and animals, the two C-4 methyl groups are removed consecutively, and a single sterol C4 methyl oxidase catalyzes the initial oxygenation reactions [8,9,10,32]. The *S. cerevisiae ERG25*-null mutants are ergosterol auxotrophs and accumulate C4,4-dimethylzymosterol [8]. Unlike *S. cerevisiae*, *Aspergillus fumigatus* has two functional sterol C4 methyl oxidases (ERG25A and ERG25B), with ERG25A being the dominant enzyme. ERG25A mutants are viable but show susceptibility to hypoxia and ER stress [33]. *Candida albicans* also has two sterol C4 methyl oxidase genes and both appear to be essential, suggesting that they work in tandem for efficient activity [23]. Similarly, plant cells possess two sterol C4 methyl oxidases, each removing one C4-methyl group in a subsequent but biochemically distinct mechanism [11,24,25]. In *Arabidopsis thaliana*, removal of each of the two C4-methyl groups requires a different sterol C4 methyl oxidase and mutants of the individual enzymes primarily accumulate their respective sterol intermediates [33]. Finally, the methanotrophic Gram-negative bacterium *Methylococcus capsulatus* contains two C-4 demethylases that are distinct from the eukaryotic enzymes [34]. Overall, these studies demonstrate diverse mechanisms involved in the C-4 demethylation step in different organisms.

In addition to *22A1*, there are other genes in *Leishmania* parasites encoding proteins homologous to *S. cerevisiae* sterol C4 methyl oxidase: the previously characterized lathosterol oxidase (LSO, LmjF.23.1300) and a putative ERG25 (LmjF.36.2540). These orthologs show 29% and 26% identity to *S. cerevisiae* sterol C4 methyl oxidase, respectively. *L. major LSO*-null mutants synthesize sterols without the C-5-C-6 double bond but show no defects in C4-demethylation [35]. In this study, we characterized the putative ERG25 in *L. major*. Like other sterol biosynthetic enzymes, ERG25 is largely localized in the ER. *L. major* ERG25-null mutants were fully viable with growth rate and virulence similar to WT parasites (Figure 4). These mutants synthesize ergosterol and 5-dehydroepisterol as their main sterols and their overall sterol composition is very similar to WT and add-back parasites (Figure 5). These observations suggest that ERG25 is not involved in the oxidative removal of C4-methyl groups. Unlike other sterol biosynthetic enzymes, the deletion or overexpression of ERG25 had no impact on the cellular levels or localization of LPG and GP63 [6,19,35].

While the precise function of ERG25 in *Leishmania* awaits further investigation, it is important to note that ERG25 is not ubiquitous among trypanosomatids. According to Tritrypdb, ERG25 orthologs are present in *Leishmania* spp., *Angomonas deanei*, *Bodo saltans*, *Crithidia fasciculata*, *Endotrypanum monterogeii*, *Leptomonas* spp., *Paratrypanosoma confusum* and *Porcisia hertigi*, but absent in *Trypanosoma* spp. In contrast, CYP5122A1 orthologs are identified in all of the above organisms including *Trypanosoma* species like *T. brucei* and *T. cruzi* [16]. Given the dispensable nature of *ERG25* in *L. major* sterol synthesis, it is possible that *ERG25* has evolved to diverge from the sterol C4 methyl oxidase function and was lost in some trypanosomatid species. Meanwhile, CYP5122A1 was conserved by most trypanosomatids as the only or dominant sterol C4 methyl oxidase.

## 4. Materials and Methods

### 4.1. Materials

M199 media and cholesta-3,5-diene were purchased from Sigma-Aldrich (St. Louis, MO, USA). Ketoconazole and posaconazole were purchased from TCI Chemicals (Portland, OR, USA) and Carbosynth (San Diego, CA, USA), respectively. DB766 was synthesized as previously described [22]. Rat anti-LdCYP5122A1 and anti-LdC14DM (CYP51) polyclonal antisera were generated at the University of Kansas Medical Center by Dr. Jianming Qiu using purified recombinant proteins as antigens. All other chemicals and instruments were purchased from Thermo Fisher Scientific (Waltham, MA, USA) unless otherwise specified.

### 4.2. Molecular Constructs

The predicted open reading frames (ORFs) of *L. major ERG25* (LmjF.36.2540) and *CYP5122A1* (LmjF.27.0090) were amplified from *L. major* LV39 (Rho/SU/59/P) genomic DNA by PCR and digested with restriction enzymes. The *ERG25* ORF was cloned into a *Leishmania* expression vector pXG to generate pXG-*ERG25* (used in add-back) or a GFP-tagged expression vector pXG-GFP2+’ to generate pXG-GFP-*ERG25* for localization studies. The *CYP5122A1* ORF (referred to as *22A1*) was cloned into the pXNG4 vector or pXG-‘GFP+ vector to generate pXNG4-*22A1* or pXG-*22A1*-GFP, respectively. All molecular constructs were verified by restriction enzyme digestion and Sanger sequencing.

To generate knockout constructs, ~600 bp of the 5′- and 3′-flanking sequences of *ERG25* ORF were PCR-amplified and cloned into the pUC18 vector, followed by the insertion of drug resistance genes for puromycin (*PAC*) or hygromycin B (*HYG*) to generate pUC18-KO-*ERG25::PAC* or pUC18-KO-*ERG25::HYG*, respectively. Similar knockout constructs for *22A1* were generated, which include ~1000 bp of the 5′- and 3′-flanking sequences and drug resistance genes for blasticidin (*BSD*) or *HYG*. Oligonucleotides used in this study are summarized in Appendix A.

### 4.3. Leishmania Promastigote Culture and Growth Rate Measurement

*L. major* strain LV39 clone 5 promastigotes were cultivated in M199 media (pH7.4) with 10% fetal bovine serum and other supplements at 27 °C, as previously described [36]. To determine growth rates, log-phase parasites (1.0–10.0 × 10^6^ cells/mL) were inoculated at 1.0–2.0 × 10^5^ cells/mL and cell densities were determined daily using a hemocytometer. During the stationary phase (2.0–4.0 × 10^7^ cells/mL), the percentage of round cells whose long axis was less than twice that of the short axis was measured by microscopy, and the percentage of dead cells was assessed by flow cytometry after propidium iodide staining, as previously described [37].

To measure the antileishmanial activity of sterol synthesis inhibitors (prepared as 10 mM stocks in DMSO), log-phase promastigotes were inoculated in complete M199 media at 2.0 × 10^5^ cells/mL in 24-well plates (1 mL/well). Inhibitors were added to various concentrations and control wells contained DMSO only (0.1–0.5%). Culture densities were determined after 48 h using a Beckman Z2 Cell Counter (Beckman Coulter, Inc., Indianapolis, IN, USA).

### 4.4. Leishmania Genetic Manipulation and Southern Blot

To delete the endogenous *ERG25* alleles, pUC18-KO-*ERG25::PAC* and pUC18-KO-*ERG25::HYG* were digested with EcoRI and HindIII to generate linear DNA fragments, which were sequentially introduced into *L. major* wild-type (WT) promastigotes by electroporation. The resulting *ERG25*-null mutant *erg25*^−/−^ (*ΔERG25::PAC/ΔERG25::HYG*) was selected based on resistance to puromycin and hygromycin. The replacement of *ERG25* alleles with *HYG* and *PAC* in *erg25*^−/−^ was confirmed by Southern blot using ^32^P-labeled probes for the ORF and an upstream flanking region of *ERG25* [38]. The add-back line *erg25*^−/−^ +ERG25 was generated by introducing pXG-ERG25 into *erg25*^−/−^. A similar approach was used to produce the heterozygous *22A1* mutant *22A1*^+/−^ (*Δ22A1::BSD/22A1*). To acquire the chromosomal *22A1*-null mutant, *22A1*^+/−^ parasites were transfected with pXNG4-*22A1* and then the second chromosomal *22A1* allele was replaced with *HYG*. The resulting chromosomal *22A1*-null mutant was confirmed by Southern blot and referred to as *22A1*^−/−^ +pXNG4-*22A1.*

### 4.5. Sterol Analysis by Gas Chromatography and Mass Spectrometry (GC-MS) and Liquid Chromatograph Tandem Mass Spectrometry (LC-MS/MS)

For GC-MS analysis, total lipids were extracted from log-phase promastigotes using the Folch method [39]. An internal standard, cholesta-3,5-diene (FW = 368.84), was added to cell lysate at 2.0 × 10^7^ molecules/cell during lipid extraction. Lipid samples were dissolved in methanol at 1.0 × 10^9^ cells/mL. Electron ionization GC-MS analyses of sterols and related lipids were performed as previously described [6]. For LC-MS/MS analysis, free sterols were isolated using a modified hexane/isopropanol extraction method and analyzed by high-performance liquid chromatography–tandem mass spectrometry (HPLC-MS/MS) using a triple quadrupole mass spectrometer as described previously [40].

### 4.6. Immunofluorescence Microscopy and Western Blot

As previously described [19], log-phase promastigotes were loaded onto poly-lysine-coated coverslips by centrifugation and fixed with 3.7% paraformaldehyde. Cells were then permeabilized with ice-cold ethanol followed by rehydration with phosphate-buffered saline (PBS). To stain the endoplasmic reticulum (ER), cells were labeled with a rabbit anti-*Trypanosoma brucei* BiP antiserum (1:2000) [17], followed by goat anti-rabbit IgG-Texas Red antibody (1:2000). To label LPG or GP63, cells were stained with mouse anti-LPG antibody WIC79.3 (1:2000) [41] or mouse anti-GP63 antibody 96/26 (1:2000), followed by goat anti-mouse IgG-FITC (1:1000). Hoechst 33342 (2.5 μg/mL) was used to stain DNA. Fluorescence microscopy was performed using a BX51 fluorescence microscope or FV3000 laser scanning confocal microscope (both made by Olympus Microscopes, Center Valley, PA, USA).

For Western blots, whole cell lysates were resolved by SDS-PAGE and transferred to polyvinylidene difluoride membranes. Blots were probed with rabbit anti-GFP (ab6556, 1:1000) (from Abcam Inc., Waltham, MA, USA, rat anti-LdCYP5122A1 (1:500), rat anti-LdCYP51 (1:500), or mouse anti-tubulin antisera (1:1000), followed by appropriate HRP-conjugated secondary antibodies (1:2000). Signals were detected using a ChemiDoc imaging system (Bio-Rad, Hercules, CA, USA).

### 4.7. GCV Treatment of Promastigotes

Next, *22A1*^+/−^ +pXNG4-*22A1* and *22A1*^−/−^ +pXNG4-*22A1* promastigotes were cultivated in the presence or absence of ganciclovir (GCV, 50 μg/mL) or nourseothricin (SAT, 150 μg/mL) and re-inoculated into fresh media containing the same negative or positive selection agents at 1.0 × 10^5^ cells/mL every 3 days [13]. The retention of pXNG4-*22A1* plasmid was assessed by measuring the level of GFP fluorescence on the third day of each passage using an Attune NxT Acoustic Flow Cytometer (Thermo Fisher Scientific, Waltham, MA, USA). After six passages, individual clones of *22A1*^−/−^ + pXNG4-*22A1* were isolated by serial dilution in 96-well plates and expanded in the presence of GCV. The GFP levels of selected clones were determined by flow cytometry.

### 4.8. Mouse Footpad Infections

After confirmation by Southern blot, all *Leishmania* mutant and add-back cells were passed through BALB/c mice to recover low passage promastigotes (<5) for virulence experiments. To assess parasite virulence, 1.0 × 10^6^ stationary phase promastigotes (day 3) were resuspended in 50 µL of DMEM and injected into the footpad of an 8-week-old female BALB/c mouse (5 mice per group). The development of footpad lesions was measured weekly using a Vernier Caliper by comparing the infected footpad with the uninfected one. Parasite numbers in infected footpads were estimated by the limiting-dilution assay [42]. During the footpad injection and measurement, an isoflurane chamber was used to anesthetize mice. Mice euthanasia was achieved through a controlled flow of CO_2_ asphyxiation.

To determine whether 22A1 is required during the amastigote stage, one half of the infected mice received GCV at 7.5 mg/kg/day for 14 consecutive days (0.5 mL each, intraperitoneal injection), while the other half received an equivalent volume of sterile PBS [43]. Mice were euthanized at the indicated time points and genomic DNA was extracted from footpads for quantitative PCR (qPCR) analyses described as follows.

### 4.9. qPCR Analyses

To determine parasite numbers in infected footpads, qPCR reactions were run in triplicates using primers targeting the 28S rRNA gene of *L. major* [43]. Cycle threshold (Ct) values were determined from melt curve analysis. A standard curve of Ct values was generated using serially diluted genomic DNA samples from *L. major* promastigotes (from 0.1 cell/reaction to 10^5^ cells/reaction) and Ct values >30 were considered negative. Parasite numbers in infected footpads were calculated using the standard curve. Control reactions included sterile water and DNA extracted from uninfected mouse footpads.

To determine pXNG4-*22A1* plasmid copy numbers in promastigotes and amastigotes, a similar standard curve was generated using serially diluted pXNG4-*22A1* plasmid DNA (from 0.1 copy/reaction to 10^5^ copies/reaction) and primers targeting the *GFP* gene. qPCR was performed using the same set of primers on DNA samples from promastigotes or infected footpads and the average plasmid copy number per cell was determined by dividing the total plasmid copy number by the total parasite number based on Ct values [43].

### 4.10. Statistical Analyses

All experiments were repeated at least 3 times and error bars represent standard deviations. Graphs were made using Sigmaplot 11.0 (Systat Software Inc, San Jose, CA, USA). Differences between experimental groups were evaluated using the Student’s *t* test (for two groups) or one-way ANOVA test (for three or more groups). *p* values indicating statistical significance were grouped into values of <0.05, <0.01 and <0.001.

## Figures and Tables

**Figure 1 ijms-25-10908-f001:**
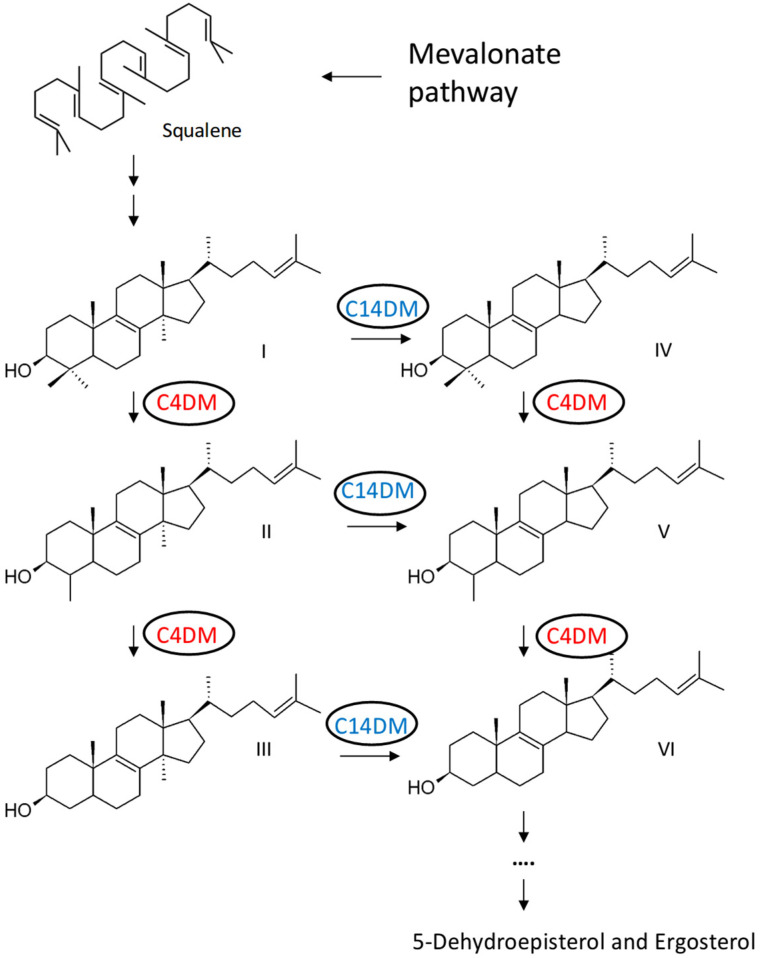
Demethylation steps catalyzed by C4DM (red) and C14DM (blue) in *Leishmania* sterol synthesis pathway. I: lanosterol; II: 4,14-dimethylzymosterol; III: 14-methylzymosterol; IV: 4,4-dimethylzymosterol; V: 4-methylzymosterol; VI: zymosterol.

**Figure 2 ijms-25-10908-f002:**
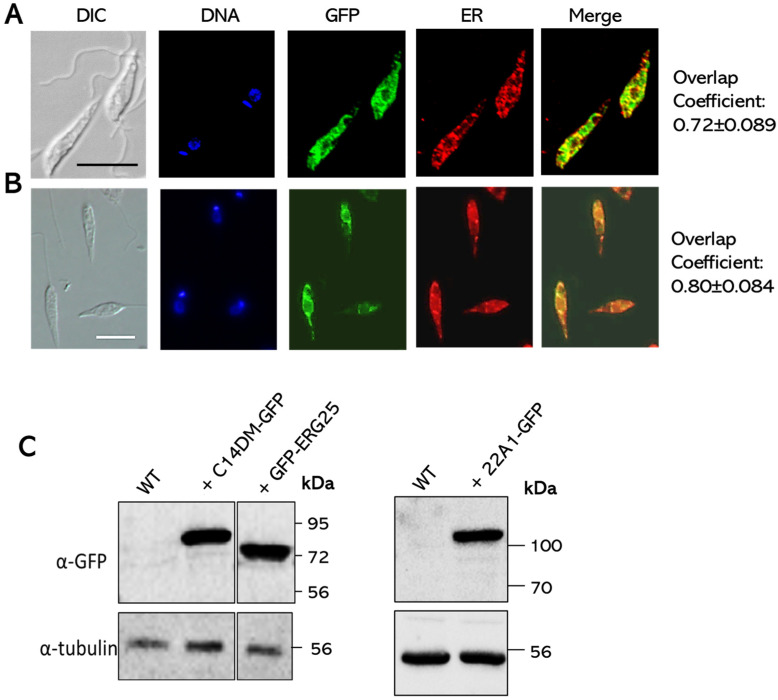
Cellular localization of CYP5122A1 and ERG25 proteins in *L. major*. (**A**,**B**) Log-phase promastigotes of WT +pXG-22A1-GFP (**A**) or WT +pXG-GFP-ERG25 (**B**) were labeled with an anti-*T. brucei* BiP antibody (ER marker) and processed for immunofluorescence microscopy. “Merge” images were the overlay of GFP and BiP. DIC: differential interference contrast. Overlap coefficients were determined from 30 cells each (average ± standard deviations). Scale bars: 10 μm. (**C**) Log-phase promastigotes of WT, WT +pXG-22A1-GFP, WT +pXG-C14DM-GFP (included to compare the sizes of GFP-ERG25 to GFP-C14DM [6]) and WT +pXG-GFP-ERG25 were analyzed by Western blot using antibodies against GFP (**top**) or α-tubulin (**bottom**).

**Figure 3 ijms-25-10908-f003:**
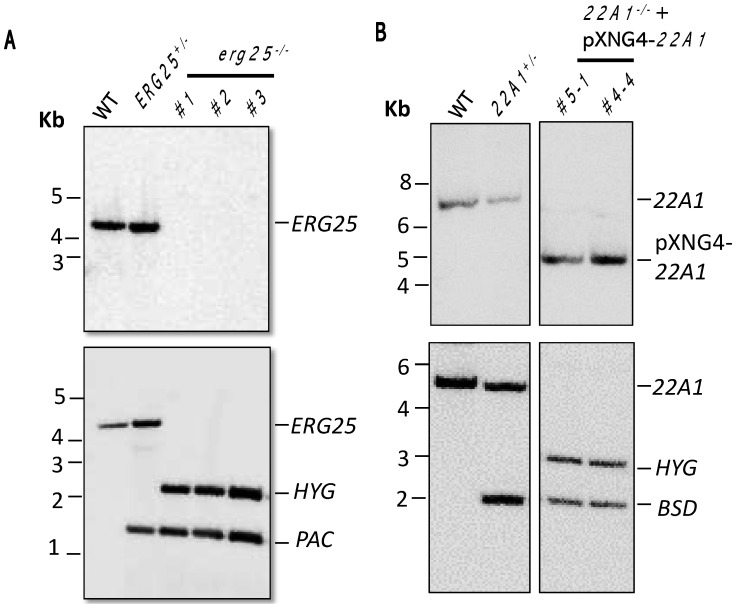
Southern blots to confirm the deletions of chromosomal *ERG25* and *22A1* alleles. Genomic DNA samples from WT, *ERG25*^+/−^, *erg25*^−/−^ (three clones), *22A1*^+/−^ and *22A1*^−/−^ +pXNG4-*22A1* (two clones) parasites were digested with restriction enzymes and hybridized with radiolabeled probes for the open reading frames (**top**) or upstream flanking regions (**bottom**) of *22A1* (**A**) or *ERG25* (**B**). Bands corresponding to *22A1*, pXNG4-*22A1*, *ERG25* and antibiotic-resistant genes (*BSD/HYG/PAC*) are indicated. Details of the Southern blots were included in Appendix A.

**Figure 4 ijms-25-10908-f004:**
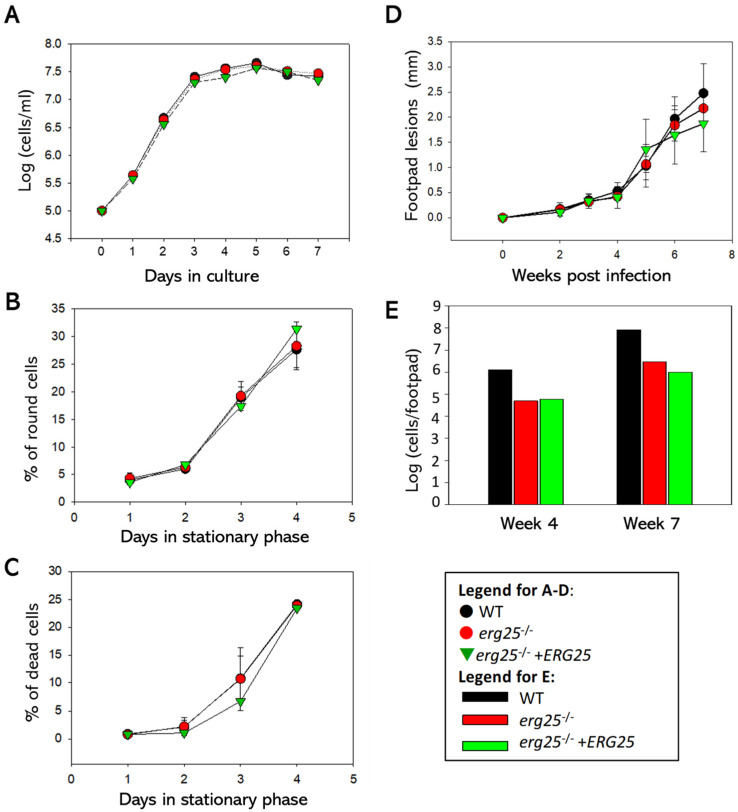
ERG25 is not required for proliferation or virulence in *L. major*. (**A**–**C**) WT, *erg25*^−/−^ and *erg25*^−/−^ +*ERG25* promastigotes were inoculated in complete M199 media at 1.0 × 10^5^ cells/mL and culture densities were determined daily (**A**). After cultures entered the stationary stage (densities > 2.0 × 10^7^ cells/mL), percentages of round cells (**B**) and dead cells (**C**) were determined daily. (**D**,**E**) Day 3 stationary phase promastigotes were injected into the footpads of female BALB/c mice. The development of footpad lesions was monitored weekly (**D**) and parasite numbers in infected footpads were determined at 4 and 7 weeks (one mouse each) post-infection via limiting dilution assay (**E**). Error bars represent standard deviations.

**Figure 5 ijms-25-10908-f005:**
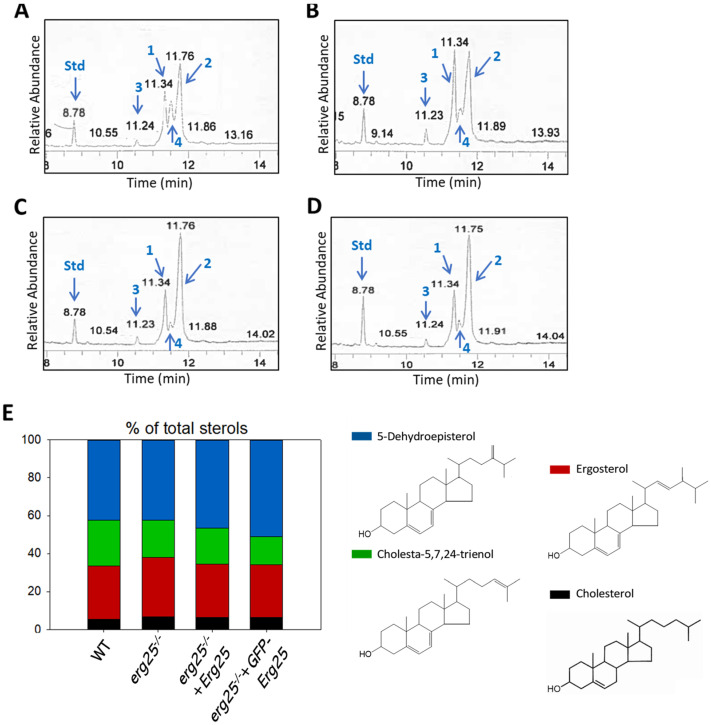
ERG25 deletion does not affect sterol composition in *L. major.* (**A**–**D**) Partial GC chromatograms of free sterols from WT (**A**), *erg25*^−/−^ (**B**), *erg25*^−/−^ +*ERG25* (**C**) and *erg25*^−/−^ +*GFP-ERG25* (**D**) promastigotes. The peaks represented ergosterol (1), 5-dehydroepisterol (2), cholesterol (3), cholesta-5,7,24-trienol (4) and cholesta-3,5-diene standard (std). Sterol compositions and structures were indicated in (**E**).

**Figure 6 ijms-25-10908-f006:**
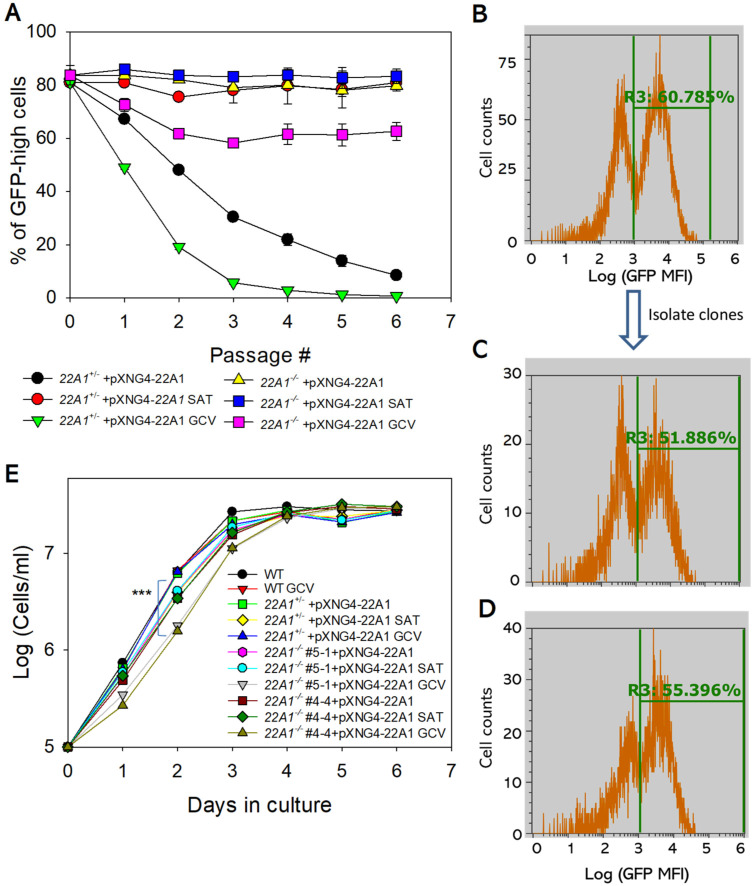
*22A1* is essential for *L. major* promastigotes in culture. (**A**) Promastigotes were continuously cultivated in the presence or absence of GCV or nourseothricin (SAT) and percentages of GFP-high cells were determined by flow cytometry for each passage. Error bars represent standard deviations from three biological replicates. (**B**–**D**) After 6 passages, single clones were isolated from *22A1*^−/−^ +pXNG4-*22A1* cells grown in the presence of GCV by serial dilution and allowed to proliferate. Flow cytometry analyses were performed on the original *22A1*^−/−^ +pXNG4-*22A1* GCV population (**B**) and two representative clones ((**C**) from GFP-low and (**D**) from GFP-high). In (**B**–**D**), percentages of GFP-high cells were indicated as R3. (**E**) Promastigotes were inoculated at 1.0 × 10^5^ cells/mL in the presence or absence of GCV or SAT and culture densities were determined daily (***: *p* < 0.001).

**Figure 7 ijms-25-10908-f007:**
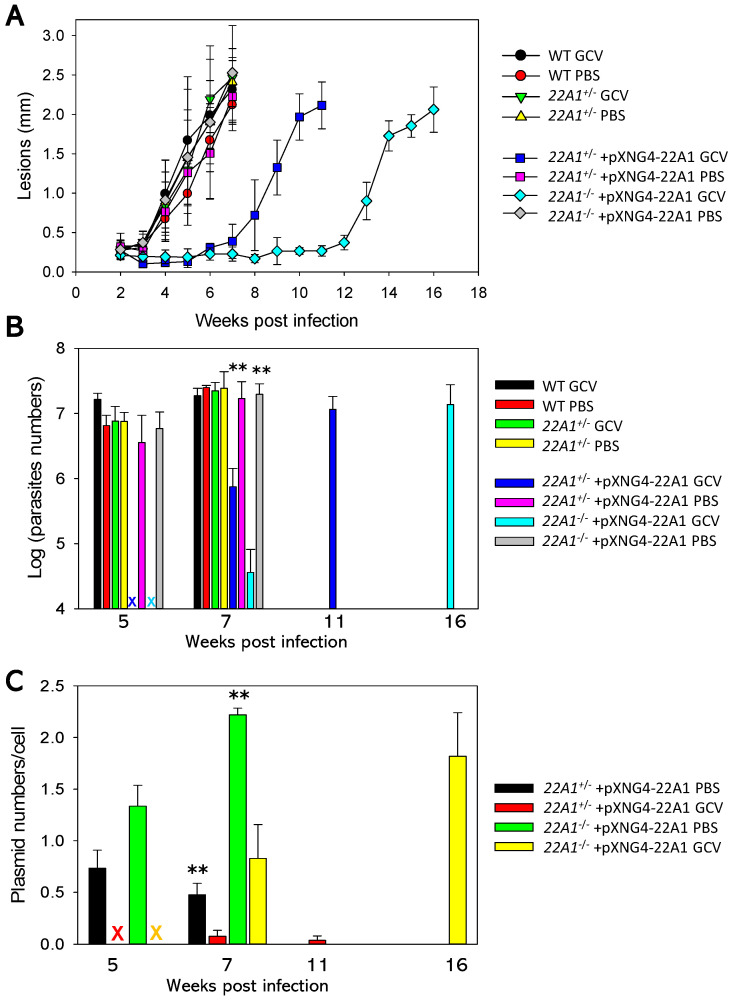
*22A1* is indispensable for *L. major* amastigotes. BALB/c mice were infected in the footpads with stationary phase promastigotes and treated with either GCV or PBS daily for 14 days. (**A**) Footpad lesions were recorded weekly. (**B**) Average parasite numbers per infected footpad were determined at the indicated times post-infection by qPCR. (**C**) Average pXNG4-*22A1* copy numbers in amastigotes were determined at the indicated times post-infection by qPCR. X in (**B**,**C**): not determined. Error bars represent standard deviations from 4–5 mice (**A**) or 2–3 mice (**B**,**C**). **: *p* < 0.01 between PBS and GCV treatment groups.

**Figure 8 ijms-25-10908-f008:**
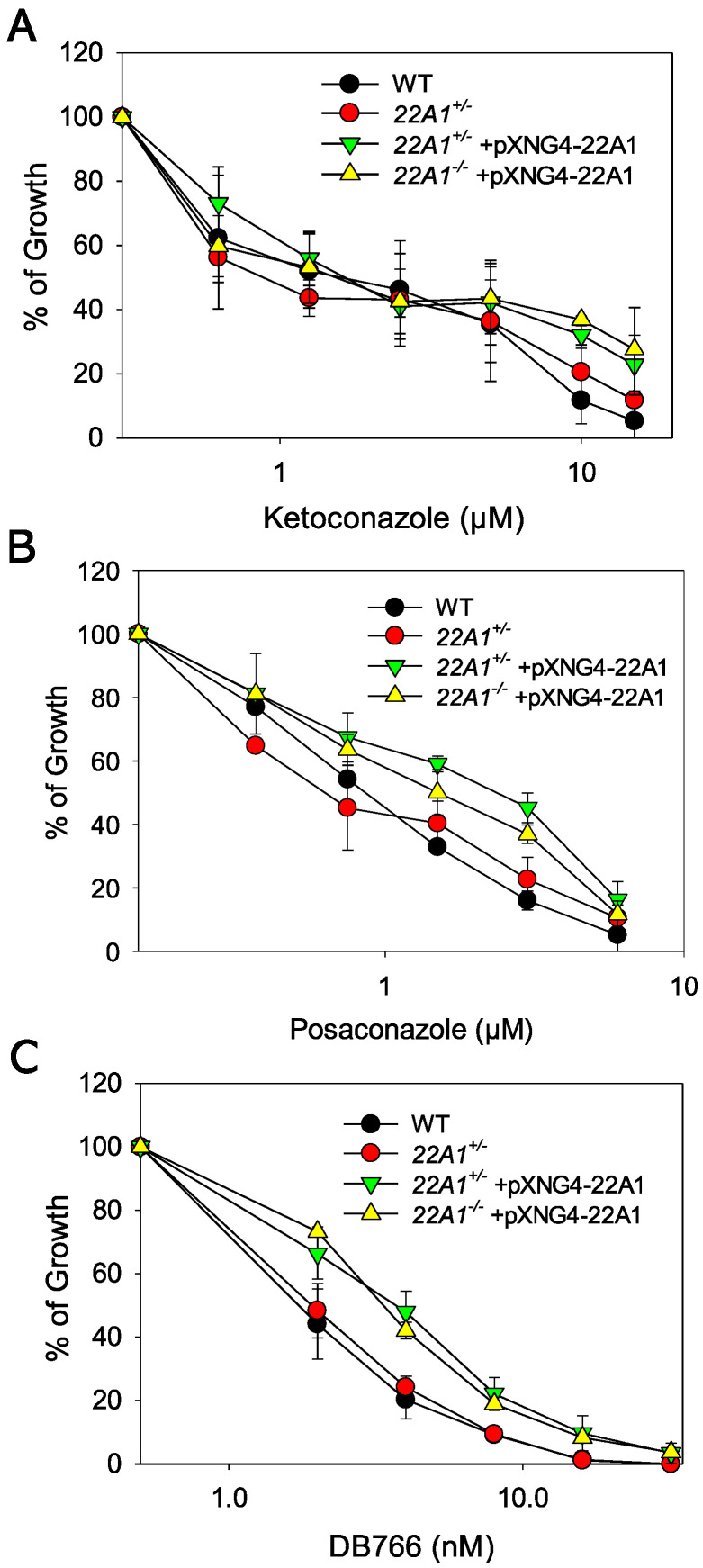
Effects of *22A1* overexpression on promastigotes’ sensitivity to sterol synthesis inhibitors. Log-phase promastigotes were inoculated in various concentrations of ketoconazole (**A**), posaconazole (**B**) or DB766 (**C**). Cell densities were determined after 48 h, and percentages of growth were calculated using cells grown in the absence of inhibitors as controls. Error bars represent standard deviations from three experiments.

**Figure 9 ijms-25-10908-f009:**
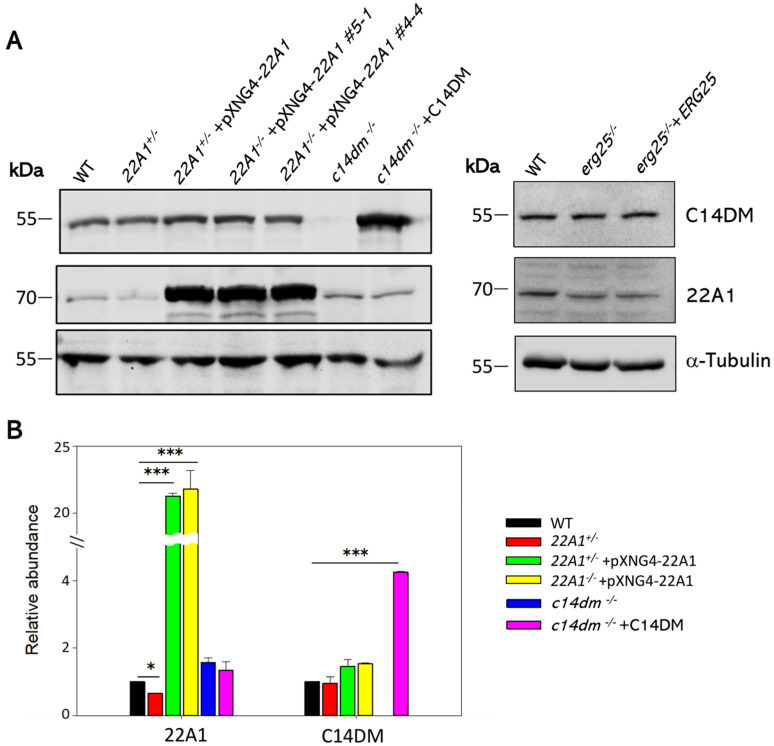
Expression levels of 22A1 and C14DM in *L. major* promastigotes. (**A**) Whole-cell lysates from log-phase promastigotes were denatured at 65 °C and analyzed by Western blot using antibodies against C14DM (top), 22A1 (middle) or α-tubulin (bottom). (**B**) The relative abundances of 22A1 and C14DM were determined after three experiments using α-tubulin as the loading control. Error bars represent standard deviations. *: *p* < 0.05, ***: *p* < 0.001.

**Table 1 ijms-25-10908-t001:** EC50 values (µM ± SD) to ketoconazole, posaconazole and DB766.

Inhibitor	WT	*22A1* ^+/−^	*22A1*^+/−^ +pXNG4-*22A1*	*22A1*^−/−^ +pXNG4-*22A1*
Ketoconazole	2.0 ± 0.055	1.0 ± 0.074 **	2.1 ± 0.14	2.0 ± 0.19
Posaconazole	1.0 ± 0.096	0.74 ± 0.18	3.0 ± 1.0 *	1.7 ± 0.38 *
DB766	1.9 ± 0.31	2.04 ± 0.41	3.8 ± 0.68 **	3.5 ± 0.19 **

Student’s *t* test was performed to analyze the difference between WT and 22A1 mutant cells (*: *p* < 0.05; **: *p* < 0.01).

## Data Availability

The original contributions presented in the study are included in the article/Appendix A, further inquiries can be directed to the corresponding author/s.

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
