# Peer review of "Molecular Characterization of Sterol C4-Methyl Oxidase in Leishmania major"

_ijms, 2024, doi:10.3390/ijms252010908_

Round 1

Reviewer 1 Report

Comments and Suggestions for Authors

Manuscript Number: IJMS-3163554

Full Title: Molecular Characterization of Sterol C4-Methyl Oxidase in 2 Leishmania major

In my opinion, the title of the manuscript is adequate. The manuscript proposes to build on the knowledge about the sterol biosynthesis pathway focusing on the characterization of C4-methyl oxidases in Leishmania major CYP5122A1, a non-canonical and C4-methyl oxidase, and a new putative C4-mehyl oxidase ERG25 ortholog present in the Leishmania genome. This was done by constructing mutants and also by their phenotypic characterization in terms of growth, infectivity and sterol metabolism.

Overall, the manuscript is well structured, the rationale is solid (although I miss some cellular data using infected cells), experiments are clear and the data is well presented considering the objectives proposed. The authors establish that: CYP5122A1 is essential in L. major (similarly to what was reported previously on Leishmania donovani) and also that the putative ERG25 ortholog has no clear observed phenotype.

 Abstract:

It is clear and showcases clearly the manuscript’s rationale, objective and results.

Introduction:

The introduction provides sufficient information to the reader for the understanding of the experiments. Some background information on ERG25 present in the discussion section (Lines381 to 386) probably would have been nice to read it in the introduction as background.

Minor notes:

Line 84 (CYP122A1 should read CYP5122A1)

Leishmania and other genus and species are sometimes not Italicized, (Ex: line 84)

Results:

On page 3 line 85 the authors report that ERG25 orthologs are not found among Trypanosoma spp without providing any immediate data or quote, (this is later done in the discussion). Probably you can quote the phylogenetic analysis done by Cosentino et al in 2014 [reference 35 of the manuscript].

In Figure 2, the presence of C14DM-GFP is not mentioned not explained in the text as relevant to this section. I think it is a control for the presence of GFP in a previous protein GFP tagged reported by the team. If so, add a reference in methods or provide background information for being in the figure. To use overexpressors for subcellular localization can be risky due the possibility of artefactual subcellular localization due to high amounts of protein (non-physiological) being produced (more than 20 copies of the plasmid are reported in Figure S6). Can the authors comment? Can the authors compare basal levels of proteins in the different mutants using antibodies for the proteins and not to GFP? This would merge nicely with the data on plasmid quantification and the EC50’s.

Why were there no in vitro infections, using cell lines or bone marrow-derived macrophages using the mutants? This information although not essential would complement the generated data for both genes. To pinpoint amastigote specific defects this would be a cleaner approach (invasion, persistence, multiplication, drug susceptibility). These experiments would be a step towards animal experiments.

The authors report (in line 136) that the mutants “appeared to have” a slower growth rate in mice (Figure 4E). The process of selection during transfection probably results in the natural attenuation of the parasites due to loss of virulence over generations. This is a well-established fact. To report this lessened virulence, or to highlight it, the authors would have to use an addback in the double KO. How do the authors manage/control this natural loss of virulence due to passages?

Just a personal note, the authors report the single KOs, as example 22A1 +/- and KO as 22A1--. I have no issue with this notation because the authors explain what each one is. Still, I found it a bit confusing, maybe it was just me. It would be more intuitive for the KO 22A1 -/- (using -/- superscript) and for the single kO use 22A1+/-. This is just a personal notation reference but woould give a more uniform notation.

Figure 7 small formatting issue in figure 7A, the legend in the right side for “22A1 +/- + pXNG4-22A1 PBS” is not at the same entry level as the others. In figure 7B and C, the representation of significance is not clear in the graphic. What bars are being compared? Are they all compared? Please, make this clearer and mention it in the text any relevant significant differences.

The negative selection approach presents some problems because it presents a phenotype visible in growth attenuation in promastigotes and reduced infections in mice. The authors attribute these differences to toxicity related to the negative selection process itself. Maybe a control using the pXNG4 plasmid empty susceptible to negative selection could have a control for this issue. The plasmid quantification experiment (figure 7c) is interesting. At first sight supports that the negative selection is working (by comparison with PBS), but the copy number is the same at endpoint with or without negative selection for the double kO, about two copies remain. Please comment on the merit of using negative selection in this experiment considering the reduced infectivity caveat and the similar endpoint data obtained in vivo.

The amount of plasmid in promastigotes was stable overtime? At the beginning of each infection was it equivalent for the sKO and dKO as shown in Figure S6? At what extent can this copy number be modulated, see my comment on the anti-parasitic activity? Still, there is striking reduction (for the double KO) to a value that would be expected for diploid organism (1 to 2 copies), it is really a nice experiment and a statement to the robustness of the approach (with or without negative selection).

Please add the Ketoconazole activity data to table 1. Concerning the drug susceptibility studies the data would profit from a non-pathway-related anti-Leishmanial drug like pentamidine or miltefosine (Miltefosine could be used but anti-parasitic activity can be modulated by this pathway). The existence of a data set together with Ketoconazole data would help solidify the effects seen for the overexpressers. Going back to the plasmid quantification and the more than 20 copies reported in Figure S6. Is it possible to think on a relationship between plasmid copy number the IC50’s, to use parasites populations with even more increased copy numbers (increasing drug pressure) to artificially increase the IC50? Why report the EC25 in table 1? It is not providing any relevant information.

Were there any drug susceptibility studies for the ERG25 mutants?

Discussion

I found the discussion is adequate for the manuscript. Just have a few questions that I would like the authors to comment.

Can ERG25 still retain the predicted biological capacity to act as a methyl oxidase but be much less abundant than CYP5122A1? Thus its activity is made redundant by the more abundant CYP5122A1. Do the authors think that ERG25 overexpression could be used for a partial rescue of CYP5122A1 KO? In folate metabolism, a similar event happens with Dihydrofolate reductase (DHFR) and Pteridine reductase 1 (PTR1) in trypanosomatids. PTR1 reduces pteridines like biopterin but is also capable of reducing folate (to a lesser extent) although this is not her major biological role and only a reduced amount of folate is actually processed by PTR1 in normal conditions. When you use anti-folate drug resistance, over time resistance can appear from overexpression of PTR1. Could the authors comment?

It is assumed that the biological importance of CYP5122A1 is due to its importance for sterol biosynthesis. Still, it cannot be excluded that it could have other biological activities that could be relevant, like the aforementioned PTR1. Moreover, some Leishmania peroxirredoxins can have unexpected roles that go beyond their predicted enzymatic activities, acting as molecular chaperones (PMID: 22046130). Were there ever any genetic studies with mutants using constructs that have no (predicted or observed) enzymatic activity to rescue the essentiality phenotype? Similar studies were done with Ribose Phosphate isomerase (PMID: 27230471), the authors used mutations that abrogated the enzymatic activity to support the importance of the biological process for the phenotype. Can the authors discuss this possibility or scenarios, if relevant?

Methods:

The section is sufficient to generically understand the experiments.

In line 467, in the western blot section there is mention to the use of “rat anti-LdCYP5122A1 (1:500), rat anti-LdCYP51 (1:500)”. What data was generated with these antibodies? The WB depicted in figure 2 is done with anti- tubulin and anti-GFP, correct?

I could not find how the EC50s in table 1 were calculated.

Author Response

Introduction:

The introduction provides sufficient information to the reader for the understanding of the experiments. Some background information on ERG25 present in the discussion section (Lines381 to 386) probably would have been nice to read it in the introduction as background.

We have moved the relevant information about ERG25 to the introduction (line 67-70).

Minor notes:

Line 84 (CYP122A1 should read CYP5122A1)

This has been corrected.

Leishmania and other genus and species are sometimes not Italicized, (Ex: line 84)

We have made the corrections.

Results:

On page 3 line 85 the authors report that ERG25 orthologs are not found among Trypanosoma spp without providing any immediate data or quote, (this is later done in the discussion). Probably you can quote the phylogenetic analysis done by Cosentino et al in 2014 [reference 35 of the manuscript].

The reference has been provided (line 70).

In Figure 2, the presence of C14DM-GFP is not mentioned not explained in the text as relevant to this section. I think it is a control for the presence of GFP in a previous protein GFP tagged reported by the team. If so, add a reference in methods or provide background information for being in the figure. To use overexpressors for subcellular localization can be risky due the possibility of artefactual subcellular localization due to high amounts of protein (non-physiological) being produced (more than 20 copies of the plasmid are reported in Figure S6). Can the authors comment? Can the authors compare basal levels of proteins in the different mutants using antibodies for the proteins and not to GFP? This would merge nicely with the data on plasmid quantification and the EC50’s.

Yes WT+C14DM-GFP was included as a control for the anti-GFP antibody and to compare the sizes of GFP-ERG25 and 22A1-GFP with C14DM-GFP. This has been clarified in Figure 3 legend with a reference.

We are aware of the possibility of overexpressed proteins generating artificial localization. However, in many cases, overexpressed proteins such as GFP-tagged chimeras have been found to accumulate in the multivesicular tubule in Leishmania for degradation (PMID: 11260523, PMID: 11260523, PMID: 11260523, PMID: 11082051). This is not the case for GFP-ERG25 or 22A1-GFP. We did not have antibody against ERG25 and we were not certain if the anti-LdCYP5122A1 antibody (generated against denatured recombinant LdCYP5122A1) would work in microscopy as it did for western blot. To alleviate the effect of overexpression, we cultivated promastigotes in the absence of G418 (the selective drug for pXG-GFP-ERG25 and pXG-22A1-GFP) for immunofluorescence microscopy. Finally, the ER localization is consistent with previous reports on ERG25 and 22A1 (PMID: 11782436, PMID: 15522820, 13, 14). Thus, we are confident about the localization results.

Why were there no in vitro infections, using cell lines or bone marrow-derived macrophages using the mutants? This information although not essential would complement the generated data for both genes. To pinpoint amastigote specific defects this would be a cleaner approach (invasion, persistence, multiplication, drug susceptibility). These experiments would be a step towards animal experiments.

In intro macrophage infection offers the advantage of quick turnaround to measure invasion and initial replication in 2-4 days, and the ease to manipulate conditions. It also has limitations as a virulence assay (short long-term only, not accounting immune response) as previously reported (PMID: 23980754). Our main goal is to determine if ERG25 is required for L. major survival and infectivity in animal models, so we went with mouse experiments directly.

The authors report (in line 136) that the mutants “appeared to have” a slower growth rate in mice (Figure 4E). The process of selection during transfection probably results in the natural attenuation of the parasites due to loss of virulence over generations. This is a well-established fact. To report this lessened virulence, or to highlight it, the authors would have to use an addback in the double KO. How do the authors manage/control this natural loss of virulence due to passages?

The ERG25 add-back was included in Figure 4 and all other assays along with the WT and ERG25-null mutants. We regularly pass WT, mutant and add-back Leishmania cells through mice to acquire low passage cells (in vitro passage <5) for virulence studies. This information is included in line 143-145 and 512-514.

Just a personal note, the authors report the single KOs, as example 22A1 +/- and KO as 22A1--. I have no issue with this notation because the authors explain what each one is. Still, I found it a bit confusing, maybe it was just me. It would be more intuitive for the KO 22A1 -/- (using -/- superscript) and for the single kO use 22A1+/-. This is just a personal notation reference but woould give a more uniform notation.

As suggested, we have changed the nomenclature for single KO (heterozygous knockout) as +/- in superscript and full KO (homozygous knockout) as -/- in superscript.

Figure 7 small formatting issue in figure 7A, the legend in the right side for “22A1 +/- + pXNG4-22A1 PBS” is not at the same entry level as the others. In figure 7B and C, the representation of significance is not clear in the graphic. What bars are being compared? Are they all compared? Please, make this clearer and mention it in the text any relevant significant differences.

The formatting issue has been fixed. The comparison in 7B and 7C is between PBS and GCV treatment groups. This has been clarified in the figure legend.

The negative selection approach presents some problems because it presents a phenotype visible in growth attenuation in promastigotes and reduced infections in mice. The authors attribute these differences to toxicity related to the negative selection process itself. Maybe a control using the pXNG4 plasmid empty susceptible to negative selection could have a control for this issue. The plasmid quantification experiment (figure 7c) is interesting. At first sight supports that the negative selection is working (by comparison with PBS), but the copy number is the same at endpoint with or without negative selection for the double kO, about two copies remain. Please comment on the merit of using negative selection in this experiment considering the reduced infectivity caveat and the similar endpoint data obtained in vivo.

22A1+/- +pXNG4-22A1 was used as a control in these experiments. GCV treatments accelerated the loss of episome in these cells under in vitro and in vivo conditions (Figure 6 and 7), indicating that the negative selection was working. GCV treatment did not have adverse effects in WT or 22A1+/- cells, thus the delayed growth/infectivity was not due to intrinsic toxicity of GCV. For 22A1-/-+pXNG4-22A1 amastigotes, GCV treatment reduced the episome copy number to ~0.8/cell (vs 2.2/cell for PBS treatment). Importantly, these numbers are much greater than those in 22A1+/- +pXNG4-22A1 GCV groups (<0.1 copies/cell), demonstrating the essentiality of 22A1 in the amastigote stage. These results were in line with other essential genes (PMID: 30926449, PMID: 37506172) but not non-essential genes (PMID: 33777852) for L. major amastigotes.  

The amount of plasmid in promastigotes was stable overtime? At the beginning of each infection was it equivalent for the sKO and dKO as shown in Figure S6? At what extent can this copy number be modulated, see my comment on the anti-parasitic activity? Still, there is striking reduction (for the double KO) to a value that would be expected for diploid organism (1 to 2 copies), it is really a nice experiment and a statement to the robustness of the approach (with or without negative selection).

In the absence of SAT, the GFP fluorescence (indicative of plasmid copy number) in 22A1+/- +pXNG4-22A1 promastigotes (Figure 6A) gradually decreased whereas the GFP fluorescence in 22A1-/- +pXNG4-22A1 promastigotes remained steady. So, at the population level, the plasmid copy number in 22A1+/- +pXNG4-22A1 is somewhat tunable.

Yes the plasmid copy numbers (20-28/cell) were determined in the 22A1+/- +pXNG4-22A1 and promastigotes prior to mouse infection. We think the high copy number contributed to the initial delay seen with 22A1+/- +pXNG4-22A1 after GCV treatment in both in intro culture (Figure 6E) and in mice (Figure 7A). The fact that 22A1+/- +pXNG4-22A1 amastigotes only retained 1-2 copies of the plasmid per cell (which differs from what we observed with the L. major CYP5122A1 study, reference [13]) suggests high 22A1 expression is harmful for L. donovani amastigotes.

Please add the Ketoconazole activity data to table 1. Concerning the drug susceptibility studies the data would profit from a non-pathway-related anti-Leishmanial drug like pentamidine or miltefosine (Miltefosine could be used but anti-parasitic activity can be modulated by this pathway). The existence of a data set together with Ketoconazole data would help solidify the effects seen for the overexpressers. Going back to the plasmid quantification and the more than 20 copies reported in Figure S6. Is it possible to think on a relationship between plasmid copy number the IC50’s, to use parasites populations with even more increased copy numbers (increasing drug pressure) to artificially increase the IC50? Why report the EC25 in table 1? It is not providing any relevant information.

EC50 for ketoconazole has been added while EC25 values have been removed from Table 1. Our previous report on L. major C14DM (CYP51) indicates that genetic or chemical inactivation of C14DM leads increased sensitivity to pentamidine (reference [7]). Studies to test inhibitor combinations for potential synergistic effects are ongoing.

Were there any drug susceptibility studies for the ERG25 mutants?

We did not perform any drug susceptibility studies on ERG25 mutants since we did not detect any abnormality in sterol composition or expression levels of CYP51/22A1 in these mutants.

Discussion

I found the discussion is adequate for the manuscript. Just have a few questions that I would like the authors to comment.

Can ERG25 still retain the predicted biological capacity to act as a methyl oxidase but be much less abundant than CYP5122A1? Thus its activity is made redundant by the more abundant CYP5122A1. Do the authors think that ERG25 overexpression could be used for a partial rescue of CYP5122A1 KO? In folate metabolism, a similar event happens with Dihydrofolate reductase (DHFR) and Pteridine reductase 1 (PTR1) in trypanosomatids. PTR1 reduces pteridines like biopterin but is also capable of reducing folate (to a lesser extent) although this is not her major biological role and only a reduced amount of folate is actually processed by PTR1 in normal conditions. When you use anti-folate drug resistance, over time resistance can appear from overexpression of PTR1. Could the authors comment?

Due to the lack of biochemical data using purified protein as described for L. major CYP5122A1 (reference [13]), we cannot rule out the possibility that ERG25 possesses some level of C4-sterol methyl oxidase activity. We did not test whether ERG25 ovexpression can bypass the lethality of 22A1 knockout, an interesting idea to follow up in the future which may also serve as an alternative to direct biochemical assay.

It is assumed that the biological importance of CYP5122A1 is due to its importance for sterol biosynthesis. Still, it cannot be excluded that it could have other biological activities that could be relevant, like the aforementioned PTR1. Moreover, some Leishmania peroxirredoxins can have unexpected roles that go beyond their predicted enzymatic activities, acting as molecular chaperones (PMID: 22046130). Were there ever any genetic studies with mutants using constructs that have no (predicted or observed) enzymatic activity to rescue the essentiality phenotype? Similar studies were done with Ribose Phosphate isomerase (PMID: 27230471), the authors used mutations that abrogated the enzymatic activity to support the importance of the biological process for the phenotype. Can the authors discuss this possibility or scenarios, if relevant?

We thank the reviewer for raising the possibility of 22A1 having a secondary function beyond sterol metabolism. No genetic studies using mutated 22A1 or a functional, heterologous ERG25 has been done to test that hypothesis.

Methods:

The section is sufficient to generically understand the experiments.

In line 467, in the western blot section there is mention to the use of “rat anti-LdCYP5122A1 (1:500), rat anti-LdCYP51 (1:500)”. What data was generated with these antibodies? The WB depicted in figure 2 is done with anti- tubulin and anti-GFP, correct?

These antibodies were used in the western blots shown in Figure 9.

I could not find how the EC50s in table 1 were calculated.

EC50s were calculated from the concentration-response curves generated using Sigma plot.

Reviewer 2 Report

Comments and Suggestions for Authors

The work shows a characterization of CYP5122A1 and ERG25 genes in Leishmania major, in order to understand the exact function of their respective translational products during the sterol synthesis. It is a very relevant topic, considering the specificity of ergosterol synthesis in these parasites, which are a crucial target for drug development. In general, the English quality and images are of excellent quality.

General comments:

The abstract is well-written, showing the main findings of the paper. I have a concern about the sentence in line 17 “..we characterized these genes in Leishmania major…” – which was exactly the characterization performed at the transcript level? I understand the importance of these genes, especially the expected essentiality for the CYP5122A1 gene was demonstrated. Still, no assays have been performed in order to really characterize the gene, like transcript stability, location, and presence in the RPN. I suggest to authors to review this term, once all characterization was performed considering the translational product of the genes. Southern and western blotting images are of outstanding quality.

Specific comments:

Line 60 – replace “half knockout” by “heterozygous”.

Line 64 – The authors reinforce the relevance of C14DM as a drug target, based on references 13 and 14. Are there any C14DM inhibitors approved or in advanced clinical studies? I wish to see more about this topic, about how relevance and important is to identify and develop new drugs for leishmaniasis treatment.

Line 65 – when and by who ERG25 was identified in Leishmania? Based on what type of assays it was shown its involvement in the demethylation process? References are missing.

Line 79 – L. major must be in italics. There are many other cases like this. Please, check out all of them.

Line 90 – it must be 2A and 2B, not C?

Figure 2 – good images, but difficult to see the scale bar. Also, in A and B, scales seem different. Why is not visible the nucleus staining at the merged image? For the localization assays, authors cloned the gene into a vector and transfected parasites, to express the tagged proteins. However, why such a process was not done by CRISPR, tagging at the original locus by inserting a myc- or an HA sequence? I have a very big concern about this strategy since overexpressor parasites are generated after the vector insertion. Checking the images, it is possible to see a high level of protein expression by this vector. How do authors ensure that the localization is the real one, and not a “side-effect” of the overexpression of these proteins? I might guess it is by the colocalization with ER but in the text, this part of the description/discussion is too shallow. Authors should improve this to guarantee the localization based on the literature.

Figure 3 – Is there any loading control to indicate that DNA was added for all samples?

Topic 2.3 – I suggest changing the terms “ERG-/+ and 22A1-/+” once they mean different things (please, I am sorry if I lost any information, but ERG-/+ is related to the EGR knockout parasites expressing ERG from a vector, generating add-back parasites, while 22A1-/+ is the heterozygote for this gene. In my opinion, the same nomenclature to refer to different things is confusing to understand. I suggest for the add-back parasites, use ERGab and -/+ for the heterozygote).

Figure 4 – please organize the images in a way that all of them have the same size. Figure E has no error bars (??). Which was the statistical analysis used to compare the groups and prove that no significant effect was observed in the biological aspects of the mutant parasites? This should be indicated in each image legend, as well as the p-value and N.

Line 134 – I have a concern about the in vivo infection using the total stationary phase parasites. We know that these parasite differentiation is quite far from those observed during the natural transmission. Is there any characterization of this cell population to prove they are mostly metacyclic? Any qPCR for a metacyclic stage-specific gene or at least a microscope image to show morphological changes? I have this concern because it is quite difficult to say that there is no difference in the infection rate without i) statistical analysis/error bars and ii) parasites in their best infective stage.

Image 5 – I suggest adding the chemical structure of each sterol analyzed side by their respective GC-MS specter.

Line 166 – It is not clear the difference between 22A1+/- and 22A1-/+. Also, it was very confusing to follow the figure 6. What is the difference between each 22A1+/- and 22A1-/+? How was the isolation of the parasites in 96well plate? Was it manual cell sorting? For how long? Why the knockout of the 22A1 was not possible without the plasmid pXNG4-22A1 but its remoting from the knockout parasites was possible? The authors should discuss this better.

Figure 7 – very nice compilation of data.

Line 341 – how is possible to relate the accumulation of squalene to the effect of 22A1 under the sterol synthesis in the parasite? Analyzing the sterol levels from parasites lacking 22A1, no difference was observed. How authors can explain this considering the fact of higher resistance to sterol synthesis inhibitors? Also, what can be used to explain that parasites lacking 22A1 are growing slower than wt, once their sterol synthesis pathway is compromised? Is there any idea? I am sorry but I was expecting some discussion about this and I could not find it.

Author Response

General comments:

The abstract is well-written, showing the main findings of the paper. I have a concern about the sentence in line 17 “..we characterized these genes in Leishmania major…” – which was exactly the characterization performed at the transcript level? I understand the importance of these genes, especially the expected essentiality for the CYP5122A1 gene was demonstrated. Still, no assays have been performed in order to really characterize the gene, like transcript stability, location, and presence in the RPN. I suggest to authors to review this term, once all characterization was performed considering the translational product of the genes. Southern and western blotting images are of outstanding quality.

We have changed that sentence to “we assessed the essentiality of these genes in Leishmania major….”.

Specific comments:

Line 60 – replace “half knockout” by “heterozygous”.

We have replaced "half knockout" with "heterozygous knockout" in the manuscript.

Line 64 – The authors reinforce the relevance of C14DM as a drug target, based on references 13 and 14. Are there any C14DM inhibitors approved or in advanced clinical studies? I wish to see more about this topic, about how relevance and important is to identify and develop new drugs for leishmaniasis treatment.

I assume the reviewer means our claim of C4DM (not C14DM which is well established) as a new antileishmanial drug target. We are in the process of developing C4DM inhibitors and C4DM-C14DM dual inhibitors. There are no specific C4DM inhibitors in advanced stages yet.

Line 65 – when and by who ERG25 was identified in Leishmania? Based on what type of assays it was shown its involvement in the demethylation process? References are missing.

The putative L. major ERG25 was annotated as a potential sterol C4-methyl oxidase by Tritrypdb and mentioned in Reference 16 which has been added. There are no functional studies for Leishmania ERG25 prior to this report.

Line 79 – L. major must be in italics. There are many other cases like this. Please, check out all of them.

Species names are italicized.

Line 90 – it must be 2A and 2B, not C?

Figure 2C shows the western blot. Figure 2A and 2B show the immunofluorescence microscopy images for 22A1-GFP and GFP-ERG25, respectively.

Figure 2 – good images, but difficult to see the scale bar. Also, in A and B, scales seem different. Why is not visible the nucleus staining at the merged image? For the localization assays, authors cloned the gene into a vector and transfected parasites, to express the tagged proteins. However, why such a process was not done by CRISPR, tagging at the original locus by inserting a myc- or an HA sequence? I have a very big concern about this strategy since overexpressor parasites are generated after the vector insertion. Checking the images, it is possible to see a high level of protein expression by this vector. How do authors ensure that the localization is the real one, and not a “side-effect” of the overexpression of these proteins? I might guess it is by the colocalization with ER but in the text, this part of the description/discussion is too shallow. Authors should improve this to guarantee the localization based on the literature.

The issue of overexpressed proteins has been addressed in our response to the other reviewer. We also like to point out that these immunofluorescence assays were done years before the CRISPR-Cas mediated methods were available. Scale bars are different in A and B because the images in A were zoomed in more than the ones in B. The merged images are overlays of GFP and BiP staining and do not include DNA staining.

Figure 3 – Is there any loading control to indicate that DNA was added for all samples?

Yes loading control (EtBr staining of DNA) was included for all DNA gels/blots in the supplemental data file.

Topic 2.3 – I suggest changing the terms “ERG-/+ and 22A1-/+” once they mean different things (please, I am sorry if I lost any information, but ERG-/+ is related to the EGR knockout parasites expressing ERG from a vector, generating add-back parasites, while 22A1-/+ is the heterozygote for this gene. In my opinion, the same nomenclature to refer to different things is confusing to understand. I suggest for the add-back parasites, use ERGab and -/+ for the heterozygote).

As described in our response to the first reviewer, we have changed the nomenclature for heterozygous knockout as +/- in superscript (+/-) and homozygous knockout as -/- in superscript (-/-). ERG25 add-back parasites are labelled as erg -/- + ERG25.

Figure 4 – please organize the images in a way that all of them have the same size. Figure E has no error bars (??). Which was the statistical analysis used to compare the groups and prove that no significant effect was observed in the biological aspects of the mutant parasites? This should be indicated in each image legend, as well as the p-value and N.

Figure 4 has been reorganized to ensure 4A-4E are of the same size. In Figure 4E, one mouse from each week (week 4 and 7) was used for the limiting dilution assay so there are no error bars in this panel. This is indicated in the legend and text. All other experiments in this study were repeated at least three time and values are presented as means ± SDs. Statistical analyses are described in the Methods section 4.10. The same method was applied to all figures.

Line 134 – I have a concern about the in vivo infection using the total stationary phase parasites. We know that these parasite differentiation is quite far from those observed during the natural transmission. Is there any characterization of this cell population to prove they are mostly metacyclic? Any qPCR for a metacyclic stage-specific gene or at least a microscope image to show morphological changes? I have this concern because it is quite difficult to say that there is no difference in the infection rate without i) statistical analysis/error bars and ii) parasites in their best infective stage.

We did measure the percentage of metacyclics in day 3 stationary phase promastigotes of WT, erg -/-, and erg -/- + ERG25 using the density centrifugation method (PMID: 11748963). No significant difference was observed as they all contained 4.4%-5.5% metacyclics. Metacyclic promastigotes were identified based on morphology.

Image 5 – I suggest adding the chemical structure of each sterol analyzed side by their respective GC-MS specter.

Chemical structures for major sterols have been added in Figure 5.

Line 166 – It is not clear the difference between 22A1+/- and 22A1-/+. Also, it was very confusing to follow the figure 6. What is the difference between each 22A1+/- and 22A1-/+? How was the isolation of the parasites in 96well plate? Was it manual cell sorting? For how long? Why the knockout of the 22A1 was not possible without the plasmid pXNG4-22A1 but its remoting from the knockout parasites was possible? The authors should discuss this better.

As mentioned above for ERG25 mutants, 22A1+/- represents heterozygous knockout and 22A1-/- +pXNG4-22A1 represents chromosomal 22A1-null mutants containing the pXNG4-22A1 plasmid. 22A1-null mutants cannot be generated without pXNG4-22A1, and the resulting 22A1-/- +pXNG4-22A1 cannot lose the plasmid, indicating that 22A1 is essential. The clone isolation in Figure 6B-D was done by serial dilution of cell culture in 92 well plates. Cells that grew at highest dilution (<1 cell/well) were considered clones and allowed to proliferate for further analysis. As shown in Figure 6C, when GFP-low clones from 22A1-/-+pXNG4-22A1 were isolated and allowed to proliferate, they showed a mix of GFP-high and GFP-low cells. This result suggests that GFP-low cells (indicative of low pXNG4-22A1 plasmid amount) cannot survive without GFP-high cells, which is consistent with the notion that 22A1 is essential.

Figure 7 – very nice compilation of data.

Line 341 – how is possible to relate the accumulation of squalene to the effect of 22A1 under the sterol synthesis in the parasite? Analyzing the sterol levels from parasites lacking 22A1, no difference was observed. How authors can explain this considering the fact of higher resistance to sterol synthesis inhibitors? Also, what can be used to explain that parasites lacking 22A1 are growing slower than wt, once their sterol synthesis pathway is compromised? Is there any idea? I am sorry but I was expecting some discussion about this and I could not find it.

We are not sure why 22A1 overexpression caused a mild squalene accumulation, which was observed in two independent 22A1+/-+pXNG4-22A1 clones (supplemental figure S9). One possibility that 22A1 overexpression caused increased metabolism of sterol intermediates leading to the upregulation of squalene synthase and/or downregulation of squalene monooxygenase (line 279-282).

Round 2

Reviewer 1 Report

Comments and Suggestions for Authors

After reading the revised version of the manuscript and reviewing the authors reposes to my concerns and doubts I consider that they adequately addressed them. Although, I still have distinct opinion on some issues raised, like the merit of using in vitro infections to address the experimental question, l accept the authors view on the subject.